# Quality of Life Identifies High-Risk Groups in Advanced Rectal Cancer Patients

**DOI:** 10.3390/healthcare13151782

**Published:** 2025-07-23

**Authors:** Anna-Lena Zollner, Daniel Blasko, Tim Fitz, Claudia Schweizer, Rainer Fietkau, Luitpold Distel

**Affiliations:** 1Department of Radiation Oncology, Universitätsklinikum Erlangen, Friedrich-Alexander-Universität Erlangen-Nürnberg (FAU), 91054 Erlangen, Germany; anna-lena.zollner@fau.de (A.-L.Z.); dblasko922@gmail.com (D.B.); tim.fitz@uk-erlangen.de (T.F.); rainer.fietkau@uk-erlangen.de (R.F.); 2Comprehensive Cancer Center Erlangen-EMN (CCC ER-EMN), Universitätsklinikum Erlangen, Friedrich-Alexander-Universität Erlangen-Nürnberg (FAU), 91054 Erlangen, Germany

**Keywords:** quality of life, QLQ-C30, advanced rectal cancer, overall survival, high-risk patients, prognosis

## Abstract

**Background/Objectives**: Quality of life (QoL) is a valuable tool for evaluating treatment outcomes and identifying patients who may benefit from early supportive interventions. This study aimed to determine whether specific QoL results in patients with advanced rectal cancer could identify groups with an unfavourable prognosis in long-term follow-up. **Methods**: A total of 570 patients with advanced rectal cancer were prospectively assessed, during and up to five years after neoadjuvant radiochemotherapy, using the QLQ-C30 and QLQ-CR38 questionnaires. We analysed 27 functional and symptom-related scores to identify associations with overall survival, once at baseline, three times during therapy, and annually from years one to five post-therapy. **Results**: Poor quality of life scores were consistently associated with shorter overall survival. The functional scores of physical functioning, role functioning, and global health, as well as the symptom scores of fatigue, dyspnoea, and chemotherapy side effects, were highly significant for overall survival at nearly all time points except for the immediate preoperative assessment. Patients over the age of 64 with lower QoL scores showed a significantly reduced probability of survival in the follow-up period, and patients who reported poor QoL in at least two of the first three questionnaires during the initial phase of treatment showed significantly reduced overall survival. **Conclusions**: Early and repeated QoL assessments, particularly within the first weeks of therapy, offer critical prognostic value in advanced rectal cancer. Identifying patients with an unfavourable prognosis might allow faster interventions that could improve survival outcomes. Integrating QoL monitoring into routine clinical practice could enhance individualised care and support risk stratification.

## 1. Introduction

Colorectal cancer (CRC) remains a substantial global health concern. It is the second leading cause of cancer-related death. With over 1.9 million new cases and almost 904,000 deaths in 2022, CRC accounted for almost 10% of all cancer cases and deaths worldwide [1]. Several studies have shown that colorectal cancer is no longer just a disease of the elderly [2,3,4,5]. Fortunately, at the same time, treatment methods have advanced substantially, which is resulting in an improved survival rate and, consequently, a growing population of CRC survivors. Therefore, focusing on these patients’ QoL can be valuable. Creating an understanding of the factors that influence their QoL is fundamental in order to improve care for survivors and implement targeted interventions to enhance their overall well-being. Patients with a higher comorbidity burden are less likely to be diagnosed via screening than via emergency cases, which are associated with a 2.8-times-higher one-year mortality rate [6]. Therefore, finding ways to support those at a higher risk after diagnosis is vital. Integrating patient-reported quality of life (QoL) assessments early in the treatment process could help provide more personalised support strategies, even when patients have missed early detection through routine screening. We assessed patient-reported QoL in a large cohort of individuals with advanced rectal cancer over a long period of time. Our primary aim was to determine whether early and repeated QoL assessments could serve as predictors of overall survival. Specifically, we investigated whether certain functional and symptom scores, measured at baseline and during the first weeks of treatment, could identify patient subgroups at higher risk of poor outcomes. Additionally, we examined how these scores evolved over time to determine if there was any prognostic value in the longer follow-up period for identifying patient subgroups within our cohort who may be at increased risk. Recognising these high-risk groups allows for more targeted support, potentially improving their long-term prognosis. Furthermore, we aimed to determine a broader yet still clinically meaningful time frame early in treatment, during which patient-reported outcomes are predictive. The idea was to detect not only other ideal points in time for assessment but also offer greater flexibility in patient assessment.

## 2. Materials and Methods

### 2.1. Study Population

This cohort study included 570 patients from the University Hospital Erlangen in Germany, consecutively recruited from May 2010 to December 2023. The patients were surveyed prospectively. The criteria for inclusion in this trial were the advanced stage of rectal cancer and neoadjuvant radiochemotherapy. Exclusion criteria were cognitive impairment and language barriers (non-German speakers). These patients, scheduled to receive radiotherapy for rectal cancer, were asked if they would like to take part in this study. In most cases, neoadjuvant radiochemotherapy was then followed by a total mesorectal excision. Figure 1 depicts the inclusion and frequency of responses, as well as the successive withdrawal of patients. The ‘lost to no response’ category indicates that these patients did not complete the questionnaires when asked twice in succession. The term ‘lost to follow-up’ refers to patients for whom the observation period was less than five years, or who could no longer be contacted. The main clinical characteristics of the surveyed patients are listed in the table below (Table 1).

### 2.2. Questionnaires

The data were collected using the third version of QLQ-C30 in conjunction with the QLQ-CR38 validated questionnaires by the European Organization for Research and Treatment of Cancer (EORTC) [7]. For each questionnaire, the validated German version was used. The EORTC QLQ-C30 consists of 30 items in total. The multi-item measures encompass five functional scales (physical, role, cognitive, emotional, and social) and three multi-item symptom scales (fatigue, pain, nausea/vomiting), as well as the single-item global health scale and a quality of life scale. Furthermore, the six most common issues reported by cancer patients (appetite loss, insomnia, constipation, dyspnoea, diarrhoea, and financial difficulties) are presented as single items. Questions on physical functioning and role functioning are answered dichotomously with “yes/no”, whereas global health status and quality of life are assessed using a linear analogue scale with a range from 1 (“very poor”) to 7 (“excellent”). The remaining items are ranked on a categorical scale from 1 (“not at all”) to 4 (“very much”). EORTC QLQ-CR38 is a colorectal cancer-specific instrument and is the recommended tool for assessing quality of life in colorectal cancer patients [8]. The 38 questions encompass four functional scales (body image, sexual functioning, sexual enjoyment and future perspective) and eight symptom scales (micturition problems, chemotherapy side effects, symptoms linked to the gastrointestinal tract, male sexual problems/female sexual problems, defecation problems, stoma-related problems, and weight loss). Subsequently, the EORTC QLQ-CR38 has been revised and updated as the EORTC QLQ-CR29 [9]. The decision in favour of the QLQ-CR38 was based on established practices at the time of study initiation in 2010. The QLQ-CR29 was validated in 2009, just prior to the start of our study [10], yet was not widely implemented. As the patients were successively recruited until 2023, the originally selected questionnaire was kept in order to ensure methodological consistency and comparability of the data over the entire survey period.

### 2.3. Data Collection Process

The study, including the use of individual patient data, was approved by the local ethics committee (Ethics Committee of the University Hospital Erlangen, approval number: 3745; date of approval: 17 April 2008). This information included the tumour stage at the time of diagnosis and details about the initial therapy, and was extracted from hospital documentation. Written informed consent for participation in the study, as well as the collection of their clinical data, was obtained from all patients. Participants under the age of 18 were included with assent and parental consent. The patients were provided with the questionnaires directly at their clinical appointment. They received the questionnaires at the beginning of their RCT (day 1), two weeks into the RCT (day 14), during week 5 at the end of the RCT (day 35), and just prior to their surgery in week 10 (day 70). From this point onwards, all of the following surveys were sent out annually by postal mail. In order for the questionnaires to be digitised, they were transferred to Excel 2016. The data were then transferred to Excel VBA (Visual Basic for Applications) and converted into 27 percentage scores (0–100%). For the functional scales, a higher percentage score means that this patient has a higher quality of life. In contrast, higher symptom scores are associated with more intense symptoms and therefore a lower QoL. In the context of this study, the term ‘lower quality of life’ is indicative of individuals who are at an elevated risk of experiencing a reduced overall survival. Although QoL data were collected over a period of up to ten years, the analysis of mean values and standard deviations was conducted for the first nine years. For the tenth year, the number of responses was too low, making meaningful statistical analysis unfeasible. The Kaplan–Meier survival analyses were limited to the first five years of follow-up. This decision was based on the aim to identify high-risk groups early in the treatment trajectory, when there is the greatest potential for timely intervention and clinical benefit.

### 2.4. Statistical Analysis

To determine optimal cut-off points in survival analyses, we used the software ‘X-tile’ (Yale University School of Medicine, Version 3.6.1), as it is a common tool for survival analysis across different medical conditions and cancer types in medical research trials [11]. This allowed us to categorise the patients into two groups, dividing them according to their scores at each observation point. To further examine differences between younger and older patients, the cohort was divided by median age (up to 64 and over 64). The threshold was used for all stratified Kaplan–Meier analyses and applied in interaction analyses to assess possible effect modification by age. After 10 years, the patients’ data were censored, meaning no further follow-up was conducted. IBM SPSS Version 29.0.2.0 (IBM Inc., Chicago, IL, USA) and GraphPad Prism (Version 10.4.1) were used for the statistical analysis, including *p*-values, means, and SD values. A Cox proportional hazards regression analysis was conducted. Variables with a *p*-value < 0.25 in the univariate analysis were included in the multivariate model. Only patients with complete data on TNM stage, age, and sex were included in the multivariable analyses. The proportional hazards assumption was assessed by inspecting the log-minus-log plots. The visualisation of overall survival was performed using Kaplan–Meier plots. Statistical significance was evaluated using the log-rank test and determined as a *p*-value < 0.05. The assumption of clinical relevance was made as soon as a difference of more than 10 percentage points was observed. This is in line with Osoba et al., who considered a difference of 10 percentage points or more to be clinically relevant [12].

## 3. Results

A total of 570 patients with advanced rectal cancer who were undergoing radiochemotherapy at Erlangen University Hospital participated in this study. The 174 female and 396 male participants, aged between 15 and 93 at diagnosis, were analysed using the two questionnaires, QLQ-C30 and QLQ-CR38. The analysis focuses on a total of nine time points, including day 1, week 2, week 5, week 10, and then on annual intervals until year 5 post-treatment. A total of 27 scores were generated from the two questionnaires, including functional and symptom-based scores. A high functional score is indicative of a positive outcome, whereas a high symptom score is indicative of a negative outcome. Conversely, a low functional score is associated with a negative outcome, and a low symptom score is associated with a positive outcome. The threshold of all scores was determined for the nine question times, and Kaplan–Meier curves were constructed. In particular, 6 out of the 27 items were found to be meaningful, as they showed a significant difference between the favourable and unfavourable patient groups at seven or more question times. These items were therefore selected for further analysis (Appendix A).

### 3.1. Detailed Evaluation of Scores

#### 3.1.1. Physical Functioning

Physical functioning was one of the most convincing scores. It describes whether patients have trouble performing strenuous activities, carrying a heavy shopping bag or a suitcase, taking a long walk or a short walk outside of the house, whether they need to stay in bed or a chair during the day, or require help with eating, dressing, washing themselves, or using the toilet. Initially, our focus was on the development of the average score throughout the observed period.

When focusing on changes in mean values over time (Figure 2A), a significant drop (−12.5 pp) in the score is recognisable between the first (76.3 ± 23.4) and the third (63.8 ± 24.2) time point. This change was recognisable in almost all scores (Appendix A). Only 2 years after treatment, the initial level is reached again and slightly exceeded (year 2: 77.1 ± 22.7). From that survey on to the 13th survey time point, physical functioning changes in a positive direction by up to 4.7 percentage points (year 9: 81.8 ± 16.1). Next, scores were used to differentiate between favourable versus unfavourable “physical functioning” for Kaplan–Meier analysis. A difference between the groups is evident in almost all data points, with the exception of week 5, where the statistical significance is not reached (*p* = 0.075) (Figure 2D). The biggest difference between the two groups occurs in year 5 (*p* < 0.001) (Figure 2J). After 10 years, 81% survived in the superior group compared to 29% in the inferior group, a difference of 52%. To further investigate the prognostic value of physical functioning, a baseline multivariate analysis was performed. Factors included were age (*p* < 0.001) and TNM-classification (T: *p* = 0.011; M: *p* < 0.001). Factors which have failed to reach statistical significance in univariate analysis were sex (*p* = 0.916) and the nodal status (*p* = 0.696). In the final multivariate analysis, the results confirmed that physical functioning is able to significantly predict overall survival (HR = 0.487, 95% CI: 0.342–0.694, *p* < 0.001). Patients with higher physical functioning scores demonstrate significantly improved survival outcomes. Subsequently, we wanted to know if this separation was related to the sex or age of the patients. Sex has little effect on the two groups, although women reported slightly worse physical functioning at baseline (−9.7 pp) and are on par with men after 6 years. Subsequent to that, however, there is a smaller increase in their level in comparison to that of the men. The differences in prognosis are similar for women and men (Appendix A). The participants’ ages were divided into under 64 s and over 64 s. The younger group deteriorated less strongly (−11.1 pp) than the older group (−14.6 pp) from the start to week 5. Until year 6, the younger group continually has a more favourable score (Figure 2K). The two age groups were divided into groups with good and poor physical functioning, and overall survival was calculated (Figure 2L,M). Up to week 10, there is no considerable disparity in physical function between the two age groups. Notably, from the fifth time point (year 1) up to year 2, a substantial gap emerges in the older group (years 1 and 2: *p* < 0.001). However, in the younger group, there was little difference between those with poor and good physical functioning. Both groups performed as well as, or even better than, the older group.

#### 3.1.2. Fatigue

The most significant symptom score is fatigue. The change in mean scores is analogous to ‘Physical functioning’ and shows a deterioration (+14.8 pp) from survey point one (day 1: 40.3 ± 27.7) to three (week 5: 55.1 ± 27.4). At the final survey point, nine years after treatment, fatigue levels improved to the lowest score of (29.3 ± 24.5) (Figure 3A). Fatigue affects the prognosis of the patients (Figure 3B–J) at all observation times, with the exception of the third observation time (*p* = 0.32) (Figure 3D). The biggest difference appears in the fourth year after treatment, with only 31% of the high-fatigue group still alive after 10 years (−46 pp) (Figure 3I). When comparing the fatigue of male and female participants, it is clear that women consistently report higher levels of fatigue, with the greatest difference observed in week 2 (−18.12 pp) (Appendix A). Fatigue scores are quite similar between age groups up until year 5. From then on, fatigue in the older patients decreases, and the difference becomes clinically significant in year 9 (Figure 3K). The survival analysis of the two age groups reveals comparable outcomes in physical functioning, with a notable distinction between age groups being evident in years 1 and 2. Higher and lower levels of fatigue significantly differ in older patients, while the younger ones show minimal disparity (Figure 3L,M). The baseline multivariate analysis for fatigue reveals that it functions as an independent predictor of overall survival (HR = 1.981, 95% CI: 1.364–2.876, *p* < 0.001).

#### 3.1.3. Global Health

The term “global health” refers to the EORTC QLQ-C30 global health status/quality of life scale. It is a summary score combining patients’ self-rated overall health and overall quality of life. For the sake of readability, we will refer to this scale as global health throughout the manuscript. Global health mean value declined from day 1 to week 5 (−10 pp), reaching a minimum of 46.2 ± 21.8. From one year post-treatment, the value remained relatively stable, attaining its maximum in year 6 at 63.7 ± 23.8 (Figure 4A). In line with the physical functioning and fatigue scores, the better and worse groups can be distinguished with statistical significance at all time points except week 5 (Figure 4A–J).

The difference between the two groups is greatest in the fifth year after cancer treatment. Patients with better global health had a survival rate of 85% at 10 years, while those with worse global health had a survival rate of only 47% (−38 pp). Patients of both age groups deteriorated towards the third time point, which corresponds to the minimum mean value (>64 years: 45.9 ± 23.1; ≤64 years: 46.5 ± 20.8) (Figure 4K). The older group’s score shows slightly better improvement from year three on than the younger group and reaches a clinically significant enhancement by the end of the survey (66.7 ± 10.5) in comparison to their initial score (56.3 ± 23.4; +10.4 pp). In contrast, younger patients report a less noticeable improvement in global health over time, with a mean score of 56.7 ± 19.9 in year nine, which is only marginally higher than on day one (56.4 ± 22.9). On day 1, the younger participants with better global health status have a clear, favourable prognosis (*p* = 0.03). Moreover, both groups, good and poor, of younger patients have a superior overall survival rate (Figure 4M). As with the previous results, a difference can be seen at year 2 post-treatment. While the older group can be divided into a statistically significant superior and inferior subgroup, this is not the case for the younger group (Figure 4L,M). In the baseline multivariate analysis, global health showed statistical significance (HR = 0.578, 95% CI: 0.357–0.936, *p* = 0.026).

#### 3.1.4. Consistent Results Across Additional Measures

Three other scores have demonstrated similarly significant results: the symptom scores for dyspnoea and chemotherapy side effects, as well as the functional score for role functioning. All of them show a deterioration in week 5 but stabilise afterwards at around the initial level. This trend is also evident in the subgrouping of the data into younger and older patients. The Kaplan–Meier analysis shows a statistically significant difference between the high- and low-scoring groups at seven to eight survey times. Week 5 is not significant, as mentioned above. From the first year following treatment, there is no discernible distinction between the favourable and unfavourable subgroups within the younger cohort (aged under 64). However, a clear separation is evident among the older patients (Appendix A).

### 3.2. Prognostic Value of Initial Questionnaires

Due to the uniform deterioration from questionnaire 1 to 3, we wanted to know if and how these first three questions affected the prognosis of the patients, depending on whether they were in the high- or low-score group. The six scores initially selected were again utilised for this analysis. Patients who were in the ‘good’ group on at least two of the three questionnaires had a significantly higher probability of survival compared to patients who scored below this threshold on at least two occasions (Figure 5A–F). Cox regression analysis has demonstrated that the three questionnaire times function as significant predictors of survival, except in the case of global health, which is close to being significant (*p* = 0.054) (Table 2 and Appendix A).

### 3.3. Switches Between the Favourable and the Unfavourable Group

We were then interested in whether and how often patients switched between favourable and unfavourable on the six significant scores. On average, each patient answered 5.8 questionnaires. A significant proportion of patients who started in the favourable group remained there. This fraction was lowest for global health at 16.4% and highest for dyspnoea at 54.5%. The proportion of those who were always unfavourable was significantly lower, at 17.3% for global health and 7.0% for dyspnoea. Among all respondents, a relatively high proportion had only one change, ranging from 16.3% for physical function to 33% for fatigue and global health. The change from favourable to unfavourable was significantly more frequent at 26.6% for global health and 27.0% for fatigue. Multiple changes of 2 or more times were observed, particularly for chemotherapy side effects (42.2%) and role functioning (46.4%). A total of 21.8% or 101 of all respondents reported an unfavourable global health score at baseline and consistently reported a favourable score thereafter. In this group of first-time negative respondents, 31.7% were women, and 46.5% were under the age of 64.5, and similarly, in all others, 30.5% were also women and 52.3% were younger than 64.5 years (Table 3).

## 4. Discussion

### 4.1. Key Findings

Our study shows that changes in functional and symptom-related QoL scores over time serve as significant predictors of long-term survival in patients with advanced rectal cancer. Specifically, poor scoring in key domains of functional and symptom scores—including physical functioning, role functioning, global health, fatigue, dyspnoea, and chemotherapy side effects—was associated with poorer prognosis. The six scores analysed in more detail were found to be independent predictors of overall survival at baseline, highlighting their prognostic value. Patients with superior scores in these domains exhibited significantly longer survival times, which reinforces the role of QoL assessments in guiding clinical decision-making. Our results highlight the vulnerability of older patients with poorer scores—particularly those over 64—who had a significantly lower likelihood of survival compared with older patients with better scores from one year post-treatment. One possible explanation for the poorer QoL scores and overall survival is that older patients are more frequently affected by chronic diseases. The study of Ricciardi et al. showed that patients with conditions, such as type 2 diabetes mellitus, obesity, or arthritis, were more likely to misattribute gastrointestinal symptoms suggestive of CRC as a side effect of their medication than as a warning sign of CRC [13]. This may delay diagnosis and consequently lead to poorer survival outcomes. QoL interviews at the start of therapy could reduce the delay. This approach may be useful for older patients or those with comorbidities, who are initially diagnosed at a later stage. Implementing tailored therapeutic approaches at an early stage may improve outcomes by reducing further physical and functional decline. A further consideration is that younger patients still have a greater potential to recover from cancer in the long term compared to older patients. Interestingly, our study did not find a significant impact of gender on health-related QoL, aligning with the findings of MS Jordhøy et al. Their research also revealed minimal differences in QoL scores between male and female patients with advanced cancer [14].

Beyond age, we identified another subgroup at increased risk: patients who, within the first three questionnaire assessments, were classified at least twice in the less favourable category. These patients consistently showed poorer survival probabilities across all examined scores. This pattern suggests that early and repeated QoL assessments could serve as a screening tool. It would enable clinicians to identify high-risk patients earlier, so they can implement more intensive supportive care strategies or even consider altered personalised treatment. In this context, digital solutions may offer an efficient and patient-friendly way to collect longitudinal QoL data. A prospective study by Schunn et al. demonstrated the feasibility and patient acceptance of an app-based approach for treatment surveillance under radiotherapy [15]. Nearly 80% of the patients completed the majority of the items [15]. Despite this potential, a recent study by Janssen et al. highlighted that the integration of digital tools into routine oncologic care is still limited [16]. Implementing technologies like this more frequently could make QoL monitoring easier in clinical practice and research, particularly in identifying high-risk patients early in their treatment phase.

Furthermore, in patients with non-small-cell lung cancer (NSCLC), the impact of high-quality nursing care was examined in a randomised controlled trial. The study’s findings indicated that patients in the high-quality nursing group demonstrated enhanced functional scores, including physical functioning and improved overall quality of life. The study also demonstrated that these patients exhibited an extended time of survival and a lower mortality rate [17]. It is also interesting to note that across all key scores, the third time point of the Kaplan–Meier analysis—in week 5 at the end of the RCT and prior to surgery—was not significant. We believe that the acute burden of side effects, which is at its maximum at the end of the RCT, could be one possible clinical factor. This burden increases for everyone, regardless of individual susceptibility. Psychological factors could also play a role, given the uncertainty that accompanies this phase. On the one hand, patients may have concerns about their response to RCT, and on the other hand, they may already be anxious about the upcoming surgery and its outcome, i.e., preoperative distress. This could lead to patients over- or underestimating themselves in this very distressing phase, resulting in increased ‘emotional homogeneity’. All these factors could potentially contribute to this temporary weakening of prognostic significance. Previous research has primarily focused on baseline QoL and its impact on patient mortality [18,19,20,21]. A key strength of our study can be seen in its extended follow-up period, which allowed us to observe long-term differences between patient groups over a five-year span post-treatment. This prolonged observation made it possible to identify age-related risk patterns. To summarise, this study demonstrates that QoL has an enduring impact on mortality rates. Moreover, early QoL assessments could be useful to provide critical prognostic insights—as shown by the results of the study—with two out of three follow-up questionnaires completed within the first five weeks of treatment.

### 4.2. Fatigue and Physical Functioning in the Literature

Several studies have established baseline physical functioning as a strong predictor of overall survival in colorectal cancer patients [18,19,20,21]. In addition, the study by Braun et al. found that a 10-point improvement in physical function at three months predicted better survival, which is in line with our findings that patients with better physical function have better survival at 10 weeks post-treatment and almost all other observation times. Our analyses also align with their results regarding global health, which demonstrated that lower global QoL is associated with higher mortality [22]. Fatigue emerged as one of the most significant symptom-related predictors of survival in our study, reinforcing the existing literature that identifies fatigue as one of the most distressing and severe symptoms in cancer patients [23,24]. A meta-analysis found cancer-related fatigue in 49% of cases, increasing to 60.6% in cases of advanced cancer [25]. These findings are consistent with our results, which indicate that fatigue remains a persistent and long-term issue. Many patients continue to experience fatigue symptoms even up to five years post-treatment—a trend that has also been documented in other studies [26,27]. As physical functioning and fatigue have a substantial impact on survival, targeted treatments could be effective for improving patient outcomes. Exercise programmes have been demonstrated to enhance physical functioning and reduce fatigue in colon cancer patients receiving chemotherapy. An 18-week supervised exercise programme has been shown to result in substantial reductions in fatigue and, in turn, to improve physical functioning when compared to the standard care [28]. Similarly, Grabenbauer et al. found that aerobic exercise (30 to 60 min per session) during and after radiotherapy and chemotherapy increased physical functioning and global health status for up to 12 months post-intervention. However, the improvement in global health was mainly observed in breast cancer patients [29]. A review by Lahart et al. on physical activity interventions in breast cancer patients also found slight improvements in health-related quality of life (HRQoL), including perceived physical functioning and fatigue. However, the review did not identify conclusive evidence regarding the impact of exercise on overall survival [30]. Likewise, a randomised controlled trial in breast cancer patients demonstrated that engaging in two hours of structured physical training per week significantly improved global health status. Although there were also small improvements in fatigue and physical functioning, the small effect sizes suggest that these promising benefits may not be substantial [31]. While these studies have shown that exercise improves physical function and reduces fatigue, it remains unclear whether these improvements are associated with an increase in overall survival. For colorectal cancer patients, there is limited evidence of a potential survival benefit from physical activity. Future studies with larger sample sizes and longer follow-up are necessary to reliably assess the long-term effects of exercising on overall survival.

### 4.3. Switches Between Favourable and Unfavourable Scores

The change analysis can indicate whether someone who starts well or badly will always remain in the respective group. This is particularly important for dyspnoea and physical function, where 54.5% and 44.4%, respectively, are always in the favourable group. It shows that people with a good physical condition and without shortness of breath have a good chance of maintaining it during and after therapy. It is pleasing that people are rarely always in the unfavourable group, where there is only a high value for fatigue, which shows that one has a good chance of getting out of an unfavourable group during therapy. There are frequent changes due to side effects of chemotherapy (42.2%), which is understandable as these have an intermittent effect. In the case of role functioning (46.4%), the change between favourable and unfavourable is the most frequent, which can be explained by the changing demands on the patient during and in the years following therapy. What is very interesting in global health is that 101 (17.7%) patients start therapy with an unfavourable score and move to the good group just by starting therapy for the first three weeks of therapy and remain there over the entire remaining observation period. We believe that this is a psychological effect caused by the fact that active treatment has finally started, and that the patient simply regains the feeling that they have a chance to regain good health.

### 4.4. Limitations

Despite the relatively large sample size of 570 patients, it should be noted that the incidence of rectal cancer is substantially higher. Our study comprises a single centre, which may limit its external validity. Therefore, future research should aim to validate the findings using multicentre or external validation cohorts to confirm the generalisability of our results. Another limitation is the subjectivity of patient-reported QoL assessments, which can be influenced by psychological, social, and contextual factors. From week 10 onward, questionnaires were mailed to patients. This could have introduced some variability, as proxies might have completed the forms and recall bias may differ from in-clinic administration. Nonetheless, patient-reported outcomes remain a valuable tool for capturing patients’ personal experiences. They have been widely validated as reliable prognostic indicators in cancer research, such as the EORTC-QLQ-C30 and QLQ-CR38 [8]. This increases the comparability and robustness of the data collected, enabling individual biases to be compensated for by the larger sample size of 570 patients. However, a potential limitation of this study is the use of the EORTC QLQ-CR38, although a revised version, the QLQ-CR29, exists and has been validated since 2009. At the start of the study in 2010, the QLQ-CR38 had been established and validated as the instrument for recording disease-specific quality of life in colorectal cancer. As the patients were recruited until 2023, the QLQ-CR38 was kept in order to ensure methodological consistency over the whole survey period. Nevertheless, the use of the older disease-specific module can lead to limitations in external comparability with more recent studies that already use the QLQ-CR29. This must be considered especially for interpretation-dependent subscales. For the main analysis of the study, five out of six QoL scores from the QLQ-C30 were used, and only one score—chemotherapy side effects—was used from the QLQ-CR38. This suggests that the impact of potential methodological limitations is minimal, as the majority of the analysed QoL data originates from the QLQ-C30. Additionally, missing data presented a challenge, particularly in later follow-up periods. Not all patients completed every questionnaire, meaning that the cohorts are inconsistent over time. It is pleasing that we have an average of 5.8 responses per patient across 570 patients. However, we did not perform imputation methods or sensitivity analyses in order to address this missing data. This may introduce bias, especially when data are not missing entirely at random. Future studies should incorporate appropriate statistical techniques to address this issue. Moreover, we analysed multiple QoL scores, which led to a large number of comparisons. False-discovery controls for multiple testing were not applied, which may increase false-positive findings. Our aim was to identify clinically meaningful trends, so we focused on six key scores: physical functioning, role functioning, global health, fatigue, dyspnoea, and chemotherapy side effects. In addition, an interaction analysis was conducted between age group and each of the six key scores for our age-stratified analyses (36 comparisons in total). Only three interactions were significant, but the overall trends were consistent, which supports the age-independent relevance of QoL in this context. The mean scores all improved over the observation period. Although it is possible that this trend partly reflects the exclusion of patients due to death, this is a common and unavoidable feature of longitudinal survival analyses. However, it is possible that some patients, who were initially classified in the less favourable group, may have experienced an improvement over time. This might have caused a switch into the more favourable group, which suggests that the observed positive trend is not solely an artefact of selective attrition but may also reflect actual recovery or more effective symptom management. Studies with improved long-term patient engagement in QoL assessments will be important for validating our results. Another limitation of our analysis is that the third assessment time point did not reach statistical significance in the multivariate Cox regression. Also, the Kaplan–Meier analysis for this time point showed no significant difference, suggesting that good or poor scores on the QLQ-C30 did not affect survival at this specific time point. This may be due to the influence of the upcoming surgical intervention, as mentioned before. Despite the lack of significance at the third time point, we kept it in our evaluation as it reflects the entire five-week period before surgery, when patients receive intensive treatment with neoadjuvant RCT. Our analysis demonstrated that patients with at least two favourable QoL scores within the initial three questionnaires showed significantly improved survival probabilities. The multivariate analysis confirmed that the first three assessments as a whole serve as independent predictors of survival. This highlights that prognostic insights are not only limited to individual time points like the baseline questionnaire but could also be obtained from other assessment time points within the first five weeks before surgery. It must be noted that the ECOG performance status, comorbidities, and the chemotherapy regimes are not included in the multivariate analysis. The reason for this is that the ECOG performance questionnaire was not conducted, comorbidity data were not regularly collected, and the chemotherapy regimen was very consistent across the patient population, so we did not expect to see substantial differences, but residual confounding cannot be fully excluded. The results are still relevant for clinicians. It provides flexibility in evaluating the patient’s prognosis even if the patient did not complete the initial questionnaire at baseline. Clinical decision-making processes and the classification of high-risk patients are thus simplified and made more accessible.

Furthermore, the majority of patients in our study were diagnosed with advanced-stage cancer, with stage 3 and 4 cases being the most prevalent. Consequently, our findings are primarily applicable to patients in this advanced stage. Lastly, the study was conducted in a German-speaking healthcare setting, and, therefore, the results are not generalisable to non-German-speaking populations. However, the findings are still highly relevant to the patient population in this setting. Future research might address this limitation by including translated questionnaire versions to capture a more diverse patient group.

### 4.5. The Relation Between Depression and Quality of Life Outcomes

In cancer patients, various factors beyond the disease can significantly impact quality of life. Cancer patients are at a higher risk of developing depression than the general population [32], which negatively affects multiple aspects of HRQoL [33]. 45% of home-care cancer patients report depressive symptoms, which in turn lead to impairments across various quality of life domains [34]. These symptoms are associated with poorer overall well-being, affecting social, emotional, and physical functioning [34], with advanced-stage patients being highly impacted [35]. Given this, it is essential to assess whether a patient reports depressive symptoms. Additionally, the study of Tung et al. has shown that fatigue at different stages of treatment impacts quality of life to such an extent that it may contribute to psychiatric symptoms [36]. Recognising and distinguishing between depressive symptoms and fatigue is therefore crucial. This issue could also be addressed in a routine screening.

### 4.6. Practical Implications for Clinical Integration

In addition to the prognostic value of our results, the incorporation of QoL surveys into routine clinical practice could also be beneficial for patients and clinicians. Early identification of high-risk patients could potentially help with deciding on an altered personalised treatment. Data has shown that patients with rectal cancer who receive radiotherapy have a higher symptom burden, such as diarrhoea and bloating, and that colorectal cancer patients who receive chemotherapy report lower global health scores [37]. Therefore, early QoL assessments could not only indicate the need for supportive care after treatment, but also help to inform treatment decisions, such as, for example, when multiple therapy options are available to patients with certain comorbidities.

Another way of using brief QoL questionnaires could be to integrate them into routine follow-up appointments, especially in settings where resources are limited. That way, patients could complete the questionnaire in the waiting room before seeing a doctor. The responses can then be evaluated using a digital scoring tool or a system that highlights problem areas, such as fatigue or constipation. This approach allows the doctor to quickly identify the specific problems and address them during the patient consultation, which makes the consultation more efficient and patient-focused. For instance, if a patient reports an increase in diarrhoea-related issues, supportive measures such as dietary counselling or medication can be initiated immediately. Moreover, the repeated interviews help to monitor the patient’s condition over time, as a decline in quality of life can be a sign of clinical deterioration. In places without many resources, these lower-cost, time-efficient tools can help to improve care without the need for additional staff or equipment.

## 5. Conclusions

Our findings support the prognostic value of QoL assessments in advanced rectal cancer. Identifying and detecting specific high-risk subgroups based on repeated QoL evaluations could increase the potential for early intervention strategies. By incorporating QoL monitoring within a more flexible time frame into routine oncological care, clinicians may be able to improve both survival outcomes and the overall well-being of patients facing advanced colorectal cancer.

## Figures and Tables

**Figure 1 healthcare-13-01782-f001:**
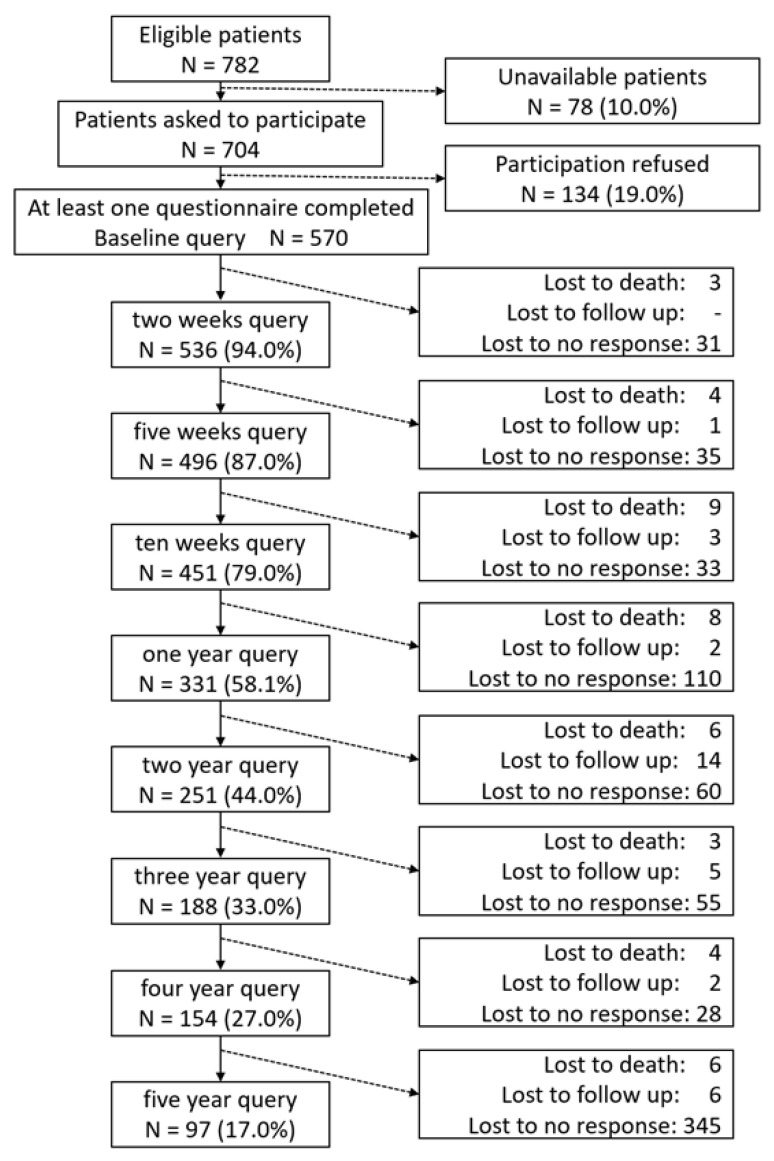
Inclusion and response behaviour, as well as the loss of 570 patients with colorectal cancer. The ‘lost to no response’ category indicates that these patients did not complete the questionnaires despite being asked twice in succession. ‘Lost to follow-up’ refers to patients for whom the observation period was less than five years or who could no longer be contacted.

**Figure 2 healthcare-13-01782-f002:**
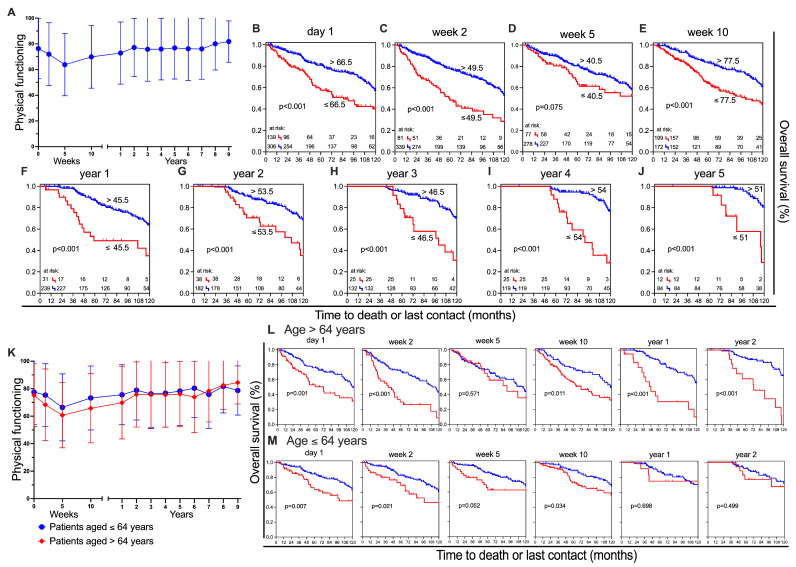
Physical functioning and overall survival. (**A**) Mean ± SD for each assessment time point in ‘physical functioning’; Kaplan–Meier survival curves for patients with higher (blue) and lower (red) scoring on physical functioning at different points in time: (**B**) day 1, (**C**) week 2, (**D**) week 5, (**E**) week 10, (**F**) year 1, (**G**) year 2, (**H**) year 3, (**I**) year 4, and (**J**) year 5. (**K**) Mean ± SD for patients stratified by age groups: >64 years (red) and ≤64 years (blue); Kaplan–Meier survival curves at six time points for patients aged >64 years (**L**) and ≤64 years (**M**), stratified by higher (blue) and lower (red) scoring for physical functioning.

**Figure 3 healthcare-13-01782-f003:**
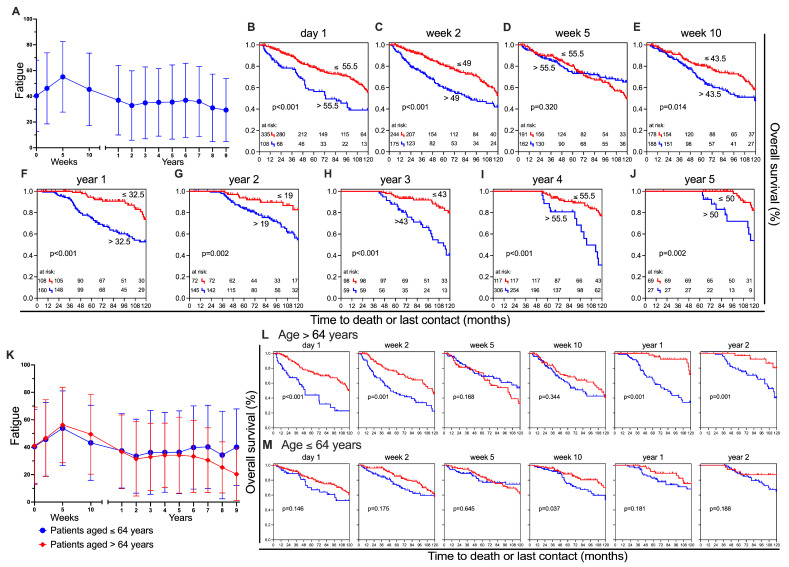
Fatigue and overall survival. (**A**) Mean ± SD for each assessment time point in ’fatigue’; Kaplan–Meier survival curves for patients with less (red) and more (blue) fatigue levels at different points in time: (**B**) day 1, (**C**) week 2, (**D**) week 5, (**E**) week 10, (**F**) year 1, (**G**) year 2, (**H**) year 3, (**I**) year 4, and (**J**) year 5; (**K**) Mean ± SD for patients stratified by age groups: >64 years (red) and ≤64 years (blue); Kaplan–Meier survival curves at six time points for patients aged >64 years (**L**) and ≤64 years (**M**), stratified by lower (red) and higher (blue) levels of fatigue.

**Figure 4 healthcare-13-01782-f004:**
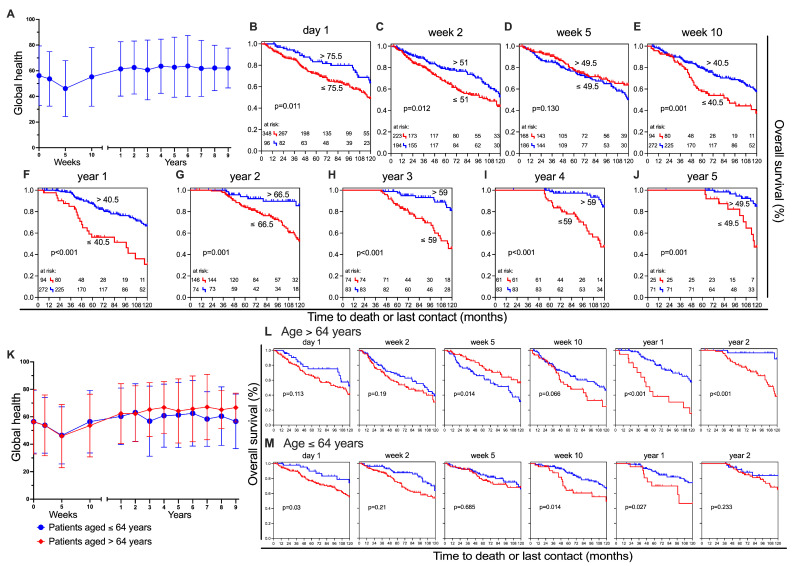
Global health and overall survival. (**A**) Mean ± SD for each assessment time point in ‘global health’; Kaplan–Meier survival curves for patients with higher (blue) and lower (red) scoring on global health at different points in time: (**B**) day 1, (**C**) week 2, (**D**) week 5, (**E**) week 10, (**F**) year 1, (**G**) year 2, (**H**) year 3, (**I**) year 4, and (**J**) year 5; (**K**) Mean ± SD for patients stratified by age groups: >64 years (red) and ≤64 years (blue); Kaplan–Meier survival curves at six time points for patients aged >64 years (**L**) and ≤64 years (**M**), stratified by higher (blue) and lower (red) scoring for global health.

**Figure 5 healthcare-13-01782-f005:**
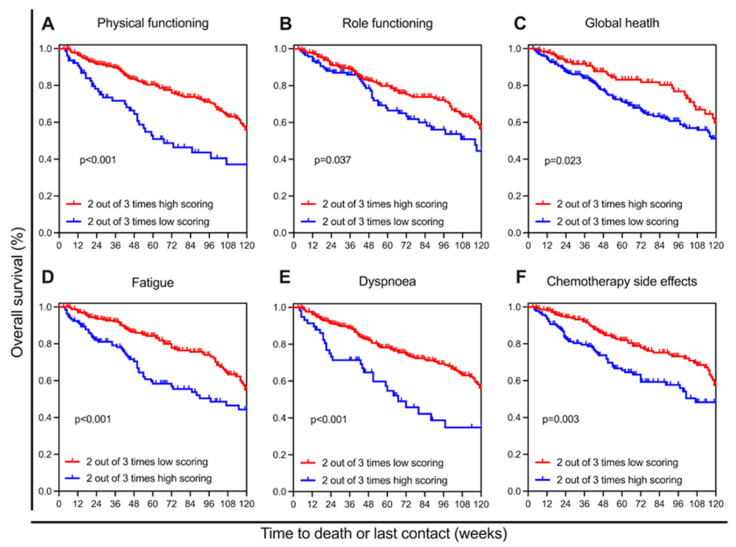
The importance of two out of three favourable/unfavourable scores from the first three surveys. Kaplan–Meier survival curves for functional scores: (**A**) physical functioning, (**B**) role functioning, (**C**) global health, and symptom scores: (**D**) fatigue, (**E**) dyspnoea, (**F**) chemotherapy side effects, red indicating patients who scored better at least twice, blue indicating those who scored worse at least twice.

**Table 1 healthcare-13-01782-t001:** Descriptive statistics of clinical characteristics of surveyed patients at diagnosis.

Characteristics	Patients (*n* = 570)
Age: mean; range; mean male; mean female (years)	62.8; 15–93; 63.0; 62.5
Sex: male; female	396 (69.5%); 174 (30.5%)
T stage: T1; T2; T3; T4; unknown	9 (1.6%); 57 (10%); 348 (61.1%); 146 (25.6%); 10 (1.8%)
N stage: N0; N1; N2; unknown	162 (28.4%); 239 (41.9%); 152 (26.7%); 17 (3%)
M stage: M0; M1; unknown	455 (79.8%); 91 (16%); 24 (4.2%)
Grade: G1; G2; G3; unknown	26 (4.6%); 437 (76.7%); 83 (14.6%); 24 (4.2%)
UICC stage: I; II; III; IV; unknown	35 (6.1%); 106 (18.6%); 311 (54.6%); 91 (16%); 27 (4.7%)

**Table 2 healthcare-13-01782-t002:** Univariate and multivariate analysis of overall survival (Cox proportional hazard model) comparing patients with at least two better scores vs. at least two poor scores.

Rectal Cancer	Univariate Analysis			Multivariate Analysis		
Variable	HR	95% C.I.	*p*	HR	95% C.I.	*p*
**Physical functioning** (poor [*n* = 77] vs. better [*n* = 321])	0.444	0.301–0.654	<0.001	0.444	0.293–0.671	<0.001
Age, years (under 64 years) [*n* = 283] vs. over 64 years [*n* = 268])	1.864	1.398–2.484	<0.001	2.365	1.605–3.485	<0.001
T category (T1/T2/3 [*n* = 410] vs. T4 [*n* = 141])	1.489	1.097–2.020	0.011	1.605	1.077–2.393	0.020
N category (N0 [*n* = 164] vs. N+ [*n* = 380])	0.941	0.692–1.279	0.696	---	---	---
M category (M0 [*n* = 412] vs. M+ [*n* = 86])	2.429	1.751–3.368	<0.001	2.644	1.737–4.023	<0.001
Gender (male [*n* = 389] vs. female [*n* = 162])	1.017	0.745–1.388	0.916	---	---	---
Variable	HR	95% C.I.	*p*	HR	95% C.I.	*p*
**Fatigue** (less [*n* = 133] vs. more [*n* = 244])	0.522	0.363–0.752	<0.001	0.560	0.379–0.829	0.004
Age, years (under 64 years) [*n* = 283] vs. over 64 years [*n* = 268])	1.864	1.398–2.484	<0.001	2.362	1.584–3.523	<0.001
T category (T1/T2/3 [*n* = 410] vs. T4 [*n* = 141])	1.489	1.097–2.020	0.011	1.408	0.922–2.151	0.113
N category (N0 [*n* = 164] vs. N+ [*n* = 380])	0.941	0.692–1.279	0.696	---	---	---
M category (M0 [*n* = 412] vs. M+ [*n* = 86])	2.429	1.751–3.368	<0.001	2.517	1.626–3.898	<0.001
Gender (male [*n* = 389] vs. female [*n* = 162])	1.017	0.745–1.388	0.916	---	---	---
Variable	HR	95% C.I.	*p*	HR	95% C.I.	*p*
**Global health** (poor [*n* = 232] vs. better [*n* = 130])	0.623	0.410–0.949	0.027	0.641	0.409–1.007	0.054
Age, years (under 64 years) [*n* = 283] vs. over 64 years [*n* = 268])	1.864	1.398–2.484	<0.001	2.576	1.698–3.907	<0.001
T category (T1/T2/3 [*n* = 410] vs. T4 [*n* = 141])	1.489	1.097–2.020	0.011	1.470	0.953–2.266	0.081
N category (N0 [*n* = 164] vs. N+ [*n* = 380])	0.941	0.692–1.279	0.696	---	---	---
M category (M0 [*n* = 412] vs. M+ [*n* = 86])	2.429	1.751–3.368	<0.001	3.141	2.025–4.873	<0.001
Gender (male [*n* = 389] vs. female [*n* = 162])	1.017	0.745–1.388	0.916	---	---	---

**Table 3 healthcare-13-01782-t003:** Switches between the more favourable and less favourable groups.

	Symptom Scores			Functional Scores		
Type of Quality of Life Answers over the Total Survey Period	Fatigue	Dyspnoea	Chemotherapy Side Effects	Physical Functioning	Role Functioning	Global Health
always favourable	104 (22.3%)	256 (54.5%)	101 (21.7%)	253 (44.4%)	90 (19.3%)	76 (16.4%)
always unfavourable	79 (16.9%)	33 (7%)	51 (11.0%)	81 (14.2%)	46 (9.9%)	80 (17.3%)
one switch, favourable to unfavourable	126 (27.0%)	73 (15.5%)	71 (15.3%)	54 (9.5%)	75 (16.1%)	123 (26.6%)
one switch, unfavourable to favourable	28 (6.0%)	35 (7.4%)	46 (9.9%)	39 (6.8%)	39 (8.4%)	29 (6.3%)
number of switches: two	74 (15.8%)	38 (8.1%)	129 (27.7%)	102 (17.9%)	146 (31.3%)	97 (21.0%)
three	37 (7.9%)	23 (4.9%)	41 (8.8%)	26 (4.6%)	42 (9.0%)	39 (8.4%)
four	14 (3.0%)	10 (2.1%)	12 (2.6%)	11 (1.9%)	19 (4.1%)	15 (3.2%)
five	5 (1.1%)	1 (0.2%)	14 (3.0%)	2 (0.4%)	8 (1.7%)	3 (0.6%)
six	0 (0%)	1 (0.2%)	0 (0.0%)	2 (0.4%)	1 (0.2%)	1 (0.2%)
two or more switches	130 (27.8%)	73 (15.5%)	196 (42.2%)	143 (25.1%)	216 (46.4%)	155 (33.5%)
first unfavourable, switch to favourable	2 (0.4%)	19 (4.0%)	8 (1.7%)	38 (6.7%)	21 (4.5%)	101 (21.8%)

## Data Availability

The data that support the findings of this study are available from the corresponding author upon reasonable request.

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
