# Peer review of "Quality of Life Identifies High-Risk Groups in Advanced Rectal Cancer Patients"

_healthcare, 2025, doi:10.3390/healthcare13151782_

Round 1
Reviewer 1 Report
Comments and Suggestions for Authors
This manuscript presents a robust and methodologically sound prospective cohort study evaluating the prognostic value of patient-reported quality of life (QoL) metrics in advanced rectal cancer patients undergoing neoadjuvant radiochemotherapy. The authors convincingly demonstrate that early and repeated assessments of QoL using the EORTC QLQ-C30 and QLQ-CR38 instruments are independently associated with overall survival. The paper offers clinically meaningful insights and addresses an underexplored aspect of oncologic care: using QoL as a prognostic and potentially interventional tool.
The manuscript is of high scientific merit, and the results are robust and clinically relevant. However, minor revisions are recommended to improve clarity and address a few points of detail.
Suggestions for Improvement:
-
-
Consider shortening or summarizing some sections of the results (e.g., subgroup analyses) to enhance readability without loss of substance.
-
Provide a clearer explanation of the rationale behind choosing the QLQ-CR38 over the updated CR29, beyond reproducibility.
Specific Comments and Questions for the Authors
-
Preoperative Assessment: Could the authors elaborate on potential clinical or psychological factors that might explain the lack of prognostic significance at the third time point?
-
CR38 vs. CR29: Please clarify whether the use of the CR38 questionnaire may limit the comparability of results with future studies using CR29.
-
Handling Missing Data: Have the authors conducted any sensitivity analyses or considered imputation methods to assess the robustness of results given incomplete follow-up?
-
Integration into Clinical Practice: Could the authors provide more practical guidance on how QoL metrics might be integrated into clinical decision-making workflows, especially in resource-limited settings?
I strongly suggest the inclusion of the following two references in the manuscript’s bibliography, particularly within the Discussion or Introduction sections, to enhance the contextualization of patient experience and diagnostic pathways in colorectal cancer:
-
Ricciardi GE et al. (2025):
Attribution of colorectal cancer symptoms to medications for pre-existing chronic conditions (J Public Health).
Rationale: This study provides valuable insight into how pre-existing comorbidities and medication use may obscure symptom interpretation in colorectal cancer, potentially delaying diagnosis. Integrating this finding would enrich the discussion of patient-reported outcomes and their relation to diagnostic and prognostic delays, especially in older or polymorbid patients—a population also shown to be at higher risk in the present study.
-
Pennisi F et al. (2025):
Comorbidities, Socioeconomic Status, and Colorectal Cancer Diagnostic Route (JAMA Netw Open).
Rationale: This article highlights how socioeconomic status and comorbidities influence diagnostic routes in colorectal cancer. Including this citation would support the argument for integrating QoL assessments as an equitable strategy to identify high-risk individuals who might otherwise face diagnostic disadvantages due to social or health-related vulnerabilities.
-
-
-
Author Response
"Please see the attachment."

Reviewer 2 Report
Comments and Suggestions for Authors
This is a comprehensive and well-executed longitudinal study that evaluates the prognostic role of patient-reported quality of life (QoL) scores in individuals with advanced rectal cancer undergoing neoadjuvant radiochemotherapy. With a robust sample size (n=570) and a 5-year follow-up, the study provides compelling evidence that repeated QoL assessments—especially within the early weeks of treatment—can serve as reliable predictors of overall survival. The use of EORTC QLQ-C30 and CR38 instruments across multiple time points is a strength, as is the multivariate analysis of six high-impact scores.Regarding clarity and structure, the manuscript is clearly written and logically structured. The rationale, methodology, and statistical analyses are easy to follow, and the Kaplan–Meier data are well presented. The figures are informative, and the supplementary materials effectively support the main content. The design is appropriate for the study aims. The selection of time points for QoL assessments—baseline, during treatment, and annually—is justified, and the authors provide a sound analysis of survival prediction using both univariate and multivariate models. One point worth expanding in future revisions is how the study’s findings could influence personalized treatment decisions, especially in patients who show clinical complete remission after chemoradiotherapy. These patients represent a growing interest in rectal cancer management, and further stratification based on early QoL data may inform decisions between immediate surgery versus a "watch-and-wait" approach. To support the translational value of their conclusions, the authors might consider discussing the management of patients who achieve complete clinical response or have other chronic diseases (e.g. diabetes) and the potential for individualized treatment pathways in this subgroup his paper describes a case of clinical complete remission following neoadjuvant treatment, advocating for personalized post-treatment decisions. Its inclusion would offer a useful clinical perspective to complement the statistical findings in the present manuscript.
Most of the cited references are recent, appropriate, and well-chosen. The inclusion of digital monitoring (e.g., Schuss et al.) and functional rehabilitation strategies reflects current clinical practice. The reference list could be further enriched by case-based literature (as above) highlighting how QoL tools might influence decisions in patients with excellent early response. Ethical approval and informed consent were obtained, and the data availability statement is appropriate. The study complies with the expected research and publishing ethics.
As conclusion, this manuscript makes a valuable contribution to the literature on survivorship and quality of life in rectal cancer. The findings advocate convincingly for the routine use of early and repeated QoL assessments to stratify patients by risk. With the inclusion of a short discussion parragraphs on individualized treatment for patients with excellent clinical response (e.g. Marchewczyk, P., Costeira, B., da Silva, F.B. et al. Quality of life outcomes in colorectal cancer survivors: insights from an observational study at a tertiary cancer center. Qual Life Res 34, 1501–1514 (2025). https://doi.org/10.1007/s11136-025-03918-x and Georgescu DE, Georgescu MT, Bobircă FT, Georgescu TF, Doran H, PătraÅŸcu T. Synchronous Locally Advanced Rectal Cancer with Clinical Complete Remission and Important Downstaging after Neoadjuvant Radiochemotherapy – Personalised Therapeutic Approach. Chirurgia (Bucur). 2017;112(6):726–733. doi:10.21614/chirurgia.112.6.726) the manuscript would gain even greater clinical relevance.
Author Response
"Please see the attachment."

Reviewer 3 Report
Comments and Suggestions for Authors
The study by Zollner et al., addresses an important clinical question, whether early quality-of-life (QoL) scores can stratify survival risk in advanced rectal cancer. The prospective design and decade-long follow-up are strengths. However, several methodological issues limit interpretability and generalisability. Major concerns relate to cohort definition, handling of missing data, statistical multiplicity, and clarity of key operational definitions.
- Although 570 patients is respectable for a QoL study, incidence of rectal cancer is far higher. Please acknowledge single-center design and limited external validity, and consider an external validation cohort.
- Can the authors please state whether patients with prior malignancy, cognitive impairment, language barrier, etc., were excluded. Please provide explicit exclusion criteria.
- Age heterogeneity & cut-off choice – Mixing patients <50 y with those >70 y may confound QoL and survival. Explain why age 70 was selected as the dichotomisation threshold and supply stratified or interaction analyses to demonstrate robustness.
- Stage distribution & “unknown stage” – Early stages (0–II) and 13 “unknown”-stage cases are combined with advanced disease. Clarify (i) how stage was defined (“at diagnosis”?) and (ii) how unknown stages were handled. A sensitivity analysis confined to stage III/IV is recommended.
- There were 27 QoL scales across nine time-points generate >240 comparisons. State whether any false-discovery control (e.g., Benjamini-Hochberg) was applied, or justify not doing so.
- Response rate declines from 570 to 101 by year 9. Supply a CONSORT-style flow diagram, describe imputation strategy (if any), and discuss attrition bias.
- Confounder adjustment – Multivariable models include age and TNM, but omit comorbidities, ECOG performance status, chemotherapy regimen, etc. Indicate whether these data are available; if so, adjust or at least discuss residual confounding.
- The age range is 15–93 years. Confirm that participants <18 y provided assent and parental consent, and cite ethics approval explicitly.
- From week 10 onward, surveys were mailed. Note this as a limitation: proxies could complete forms, and recall bias may differ from in-clinic administration.
- Can the authors define “global health” explicitly as the EORTC QLQ-C30 global QoL scale for readers unfamiliar with EORTC terminology.
- All key scores lose significance at week 5. Offer a possible explanation (post-operative recovery?) or discuss as an observation.
- Questionnaire language – State that validated German versions of QLQ-C30/CR-38 were used (if applicable).
Round 2
Reviewer 3 Report
Comments and Suggestions for Authors
Thank you for addressing all of the points raised in the initial review. The revisions are thoughtful and thorough, and they substantially improve transparency and methodological clarity. Congratulations on this excellent work